# Supervised deep learning with gene functional annotation for cell classification

Zhexiao Lin[1], Yuanyuan Gao[1], Wei Sun [2,3,4]*

1 Department of Statistics, University of California, Berkeley, California, United States of America, 2 Public Health Sciences Division, Fred Hutchinson Cancer Center, Seattle, Washington, United States of America, 3 Department of Biostatistics, University of Washington, Seattle, Washington, United States of America, 4 Department of Biostatistics, University of North Carolina, Chapel Hill, North Carolina, United States of America

* wsun@fredhutch.org

## Abstract

Gene-by-gene differential expression analysis is a widely used supervised approach for interpreting single-cell RNA-sequencing (scRNA-seq) data. However, modern scRNA-seq datasets often contain large numbers of cells, leading to the identification of many differentially expressed genes with extremely small p-values but negligible effect sizes, thus making biological interpretation difficult. To overcome this challenge, we developed Supervised Deep learning with gene functional ANnotation (SDAN), a method that integrates gene functional annotation information (e.g., protein-protein interaction) with gene-expression profiles through a graph neural network. SDAN identifies functionally coherent gene sets that optimally classify cells, and the resulting cell-level classification scores can be aggregated to make individual-level predictions. We evaluated SDAN alongside three representative existing methods in three real-data applications aimed at identifying gene sets associated with severe COVID-19, dementia, and cancer immunotherapy response. Across all applications, SDAN consistently outperformed the alternative approaches by achieving two objectives simultaneously: accurate outcome classification and clear assignment of genes to functionally related gene sets.

## Author summary

SDAN is a computational method that summarizes scRNA-seq differential expression results at the level of gene sets. These gene sets are learned to achieve two complementary goals: accurate cell classification and representation of coherent biological functions. In practice, gene sets are much easier to interpret than long lists of differentially expressed genes. We demonstrate the utility of SDAN across three real-world datasets, showing that it identifies gene sets that not only distinguish cells but also classify individuals according to clinical outcomes.

**Data availability statement:** "Su et al. COVID-19 dataset: Gene expression data were downloaded from ArrayExpress: https://www.ebi.ac.uk/biostudies/arrayexpress/studies/E-MTAB-9357 on 2/22/2022. Patient information was extracted from Table S1 of Su et al.: https://data.mendeley.com/datasets/tzydswh-hb5/5 on 2/22/2022." SEA-AD dataset: The snRNA-seq data were downloaded from cellx-gene website https://cellxgene.cziscience.com/collections/1ca90a2d-2943-483d-b678-b809b-f464c30 on 2021/11/21. Donor meta data and clinical data were downloaded from the SEA-AD website https://portal.brain-map.org/explore/seattle-alzheimers-disease/seattle-alz-heimers-disease-brain-cell-atlas-download on 2022/1/7. Cancer immunotherapy dataset: The gene expression data of Sade-Feldman et al. 2018 were downloaded from https://www.ncbi.nlm.nih.gov/geo/query/acc.cgi?ac-c=GSE120575. on 2023/11/6. Cell and sample information was extracted from Supplementary Table 1 of Sade-Feldman et al. 2018. The gene expression and meta-data of Yost et al. 2019 were downloaded from https://www.ncbi.nlm.nih.gov/geo/query/acc.cgi?acc=GSE123813. Codes for data processing and running SDAN for all the datasets are available at https://github.com/Sun-lab/SDAN.

**Funding:** This work was supported by the National Institutes of Health (HG013177 to WS; GM105785 to WS and ZL). The funders had no role in study design, data collection and analysis, decision to publish, or preparation of the manuscript.

## Introduction

A key step in the analysis of single-cell RNA-seq (scRNA-seq) data is gene-by-gene differential expression (DE) analysis, often followed by identification of biological processes enriched among the DE genes. However, because modern scRNA-seq datasets can contain very large numbers of cells, DE analyses frequently produce extremely small p-values for many genes even when the corresponding effect sizes are negligible. As a result, researchers often need to apply additional ad hoc filtering to obtain interpretable results. To address this challenge, we focus instead on directly identifying gene sets that accurately classify cells. Gene sets are typically more biologically interpretable than long lists of DE genes, and classification accuracy is often more relevant for practical applications than p-values alone.

To achieve this goal, we developed Supervised Deep Learning with gene functional ANnotation (SDAN), a graph neural network (GNN)-based method that integrates gene functional annotation information (e.g., protein–protein interactions) with gene expression data. SDAN constructs a gene–gene interaction graph to represent annotation relationships, incorporates gene expression as node features, and learns gene programs whose aggregated expression profiles classify cells and whose member genes exhibit coherent expression patterns and proximity in the graph. Unlike conventional neural network models that often function as "black boxes," SDAN produces inherently interpretable outputs in the form of gene programs, enabling transparent and biologically meaningful interpretation of the learned representations.

A central strength of deep learning is representation learning, namely the ability to derive new features that are often complex functions of the original inputs. In scRNA-seq analysis, however, the original features (genes) are already biologically meaningful and inherently interpretable, which limits the benefit of replacing them with unconstrained latent representations. This may help explain why most deep learning applications in scRNA-seq have focused on unsupervised tasks, including dimension reduction and denoising [1–3], clustering [4,5], and batch correction or harmonization [6,7], rather than supervised prediction. SDAN can be viewed as combining unsupervised grouping of genes with supervised classification of cells. The model learns gene programs that are directly optimized for classification performance while remaining interpretable and biologically coherent.

Several factorization-based methods have been developed to identify gene programs that capture shared structure in single-cell data. Although these methods are not designed for supervised classification, the gene programs they produce can be used to train downstream classifiers, allowing a fair comparison with SDAN. Broadly, factorization-based methods can be divided into those that do and do not incorporate gene functional annotations. Among methods that do not use gene functional annotations, a representative example is consensus non-negative matrix factorization (cNMF) [8]. More recently, sciRED was developed as a computationally efficient factorization-based framework for large-scale scRNA-seq data, and it outperformed alternative factorization methods (including NMF-based approaches) in terms of combined interpretability and runtime efficiency [9]. A second class of methods incorporates gene functional annotations. A representative example is Spectra [10], which

models both cell-type-specific and cell-type-agnostic factors. In addition, scNET [11] integrates scRNA-seq data with protein–protein interaction networks to learn context-specific gene and cell embeddings. We compare SDAN with sciRED, Spectra, and scNET across three real-world classification tasks: distinguishing severe from mild COVID-19, dementia from healthy controls, and cancer immunotherapy responders from non-responders. In all three settings, the ultimate goal is to classify individuals, although predictions are first generated at the cell level and then aggregated to produce individual-level predictions.

## Results

### Overview of SDAN

SDAN can be viewed as a two-step method (Fig 1). In the first step, it integrates scRNA-seq data with gene functional annotations to identify gene programs using a graph neural network (GNN). Each gene program is represented as a linear combination of genes. These gene programs are learned through a graph pooling operation, in which genes with similar expression patterns and similar annotations (i.e., genes connected in the graph) are pooled into the same gene program. To facilitate interpretation, the objective function includes a regularization term that encourages sparse loadings, making the assignments of genes to gene programs nearly binary. As a result, the learned gene programs can also be interpreted as gene sets. In the second step, the scRNA-seq data are projected onto these gene programs, and the resulting gene-program-level representation is used to classify cells with a multilayer perceptron (MLP). The two neural network components, the GNN for gene program learning and the MLP for outcome prediction, are trained jointly.

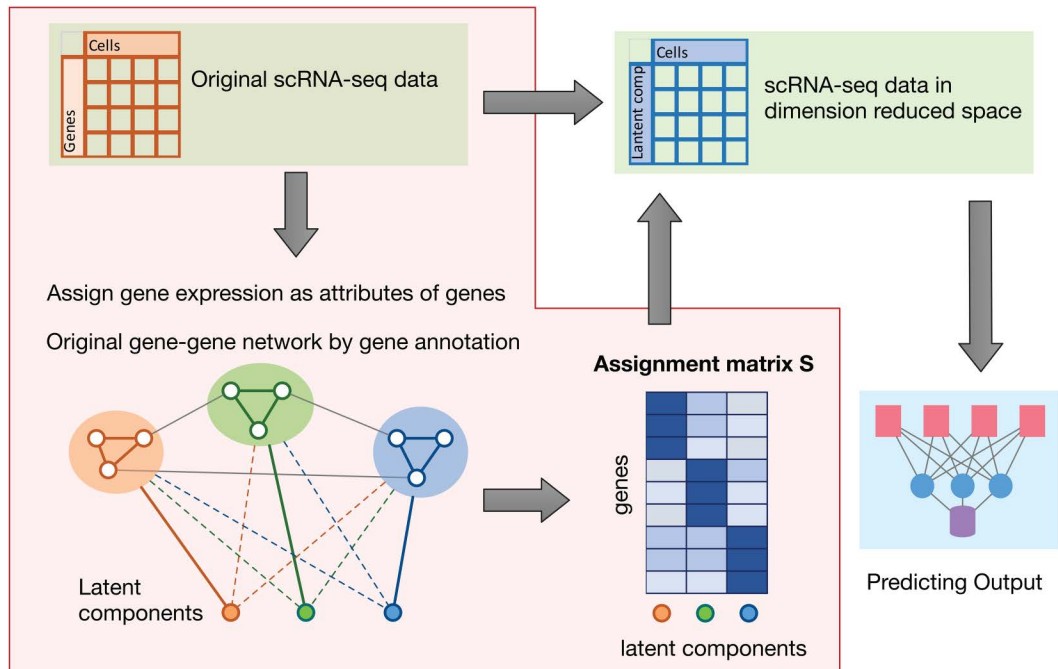

**Fig 1. Overview of Supervised Deep Learning with Gene functional Annotation (SDAN).** In the first step (shaded area), SDAN integrates gene expression data with gene functional annotations represented by a gene–gene interaction graph to learn a gene loading matrix **S**, which specifies the weights of each gene across all gene programs. In the second step, SDAN uses **S** to project each cell's gene expression profile into a low-dimensional space, reducing the representation from the number of genes to the number of gene programs, and then makes predictions in this low-dimensional space using an MLP.

The loss function of SDAN consists of an unsupervised component and a supervised component. The unsupervised loss governs the GNN and encourages genes that are connected in the graph to be grouped into the same gene program. The supervised loss is the cross-entropy loss used to optimize classification accuracy. Details are provided in the Methods section. We controlled the relative contribution of the unsupervised term by introducing a weight parameter and evaluated a range of values from 0 to 10. When this weight is set to 0, SDAN reduces to a baseline approach that does not use gene annotations. We provide practical guidance for selecting this weight and found that a value of 2 was a robust choice across all datasets analyzed, as it yielded accurate predictions together with strong functional coherence among genes within each gene set.

We emphasize that the primary goal of SDAN is to learn interpretable gene programs that are relevant to a phenotype of interest, rather than to optimize prediction accuracy alone. In SDAN, classification provides the supervisory signal that selects gene programs associated with the outcome, whereas gene annotations and sparsity constraints ensure that these gene programs remain biologically coherent and interpretable. Accordingly, the main output of SDAN is a set of phenotype-relevant gene programs, and the prediction scores should be viewed as a downstream summary derived from this gene program representation.

We have also compared SDAN with a baseline method: differential-expression followed by functional category enrichment analysis. To implement this baseline, we first identified differentially expressed genes between groups, then performed functional category enrichment analysis to identify the top 40 "baseline terms". See Section E in S1 Appendix for details. The majority of the SDAN gene programs (approximately 75%) do not have significant overlap with those baseline terms. The results remain similar when we focus on those more informative SDAN gene programs that classify the phenotype well and have high connectivity in the gene–gene interaction graph (Table D in S1 Appendix). This suggests that the baseline terms miss many informative SDAN gene programs. In addition, we also compared classification accuracy. Across five benchmark settings, SDAN outperformed the classical DE + enrichment workflow in four settings and remained competitive in the fifth setting. Details and results are provided in Section E in S1 Appendix.

## Gene expression in CD8 + /CD4 + T cells can distinguish severe from mild COVID-19 patients

It has been well established that COVID-19 vaccines can reduce the risk of severe COVID-19 disease. Vaccines induce antibodies and memory T cells to fight the SARS-CoV-2 virus. While antibodies protect us against infection [12–14], they fail to Omicron variants of SARS-CoV-2 virus [15,16]. Several recent studies have shown that a prompt T cell response protect us from severe COVID-19 [17–23]. These results, combined with the finding that vaccine-induced memory T cells cross-recognize SARS-CoV-2 variants from Alpha to Omicron [22] suggest that T cells are the key factors to prevent severe COVID-19. Therefore, we study the relationship between T-cell gene expression and COVID-19 severity.

It is worth noting that, although our focus is on cell-type-specific gene expression, cell-type proportions themselves are also informative for predicting severe COVID-19. For example, severe disease is associated with myeloid expansion (e.g., neutrophils) and lymphocyte depletion, which are likely the consequences rather than causes of the severe COVID-19 [24]. This is because an ineffective or delayed T cell response may lead to over-activation of the innate immune system, causing tissue damage and severe COVID-19. In contrast, a timely and effective T cell response can clear the virus, thereby preventing severe COVID-19.

We analyzed scRNA-seq data from ~ 42,000 CD4 + T cells and ~ 24,000 CD8 + T cells obtained from 49 patients with mild COVID-19 and 32 patients with severe COVID-19 [25]. Patients were randomly split so that half were used for training and the other half for testing. In addition, 10% of the cells in the training set were reserved as validation data for early stopping during model training. Additional preprocessing details are provided in S1 Appendix.

We first illustrate the consequence of different weights on the unsupervised loss while predicting severe COVID-19 by CD8 + T cell gene expression. Similar patterns were observed in the other applications. SDAN estimates 40 gene programs (gene sets) by default. When the weight on the unsupervised loss is small, many gene programs are empty, i.e.,

without contributing genes. As the weight increases, the number of non-empty gene programs increases and eventually stabilizes at 40 (Fig 2(A)). To assess the structural coherence of each gene program, we quantified whether the genes within a program were more connected in the annotation graph than expected by chance. For an observed gene set $G$, let $C_G$ denote its connectivity, defined as the average degree within the set, i.e., 2× (number of edges)/(number of genes). We then repeatedly sampled random gene sets of the same size and computed the empirical distribution of their connectivities, denoted by $f(G)$. The connectivity quantile of the observed gene set was defined as the quantile of $C_G$ relative to this reference distribution. As expected, increasing the weight on the unsupervised loss led to larger connectivity quantiles, indicating stronger graph coherence of the inferred gene programs (Fig 2(B)). We next aggregated cell-level prediction scores by averaging them within each patient to obtain individual-level prediction scores. The accuracy of the prediction decreased slightly as the weight on the unsupervised loss increased (Fig 2(C)). The CD4+T and CD8+T cell-level AUCs vary from 0.89 to 0.85 and from 0.91 to 0.83, respectively, whereas individual-level AUCs are more stable and are

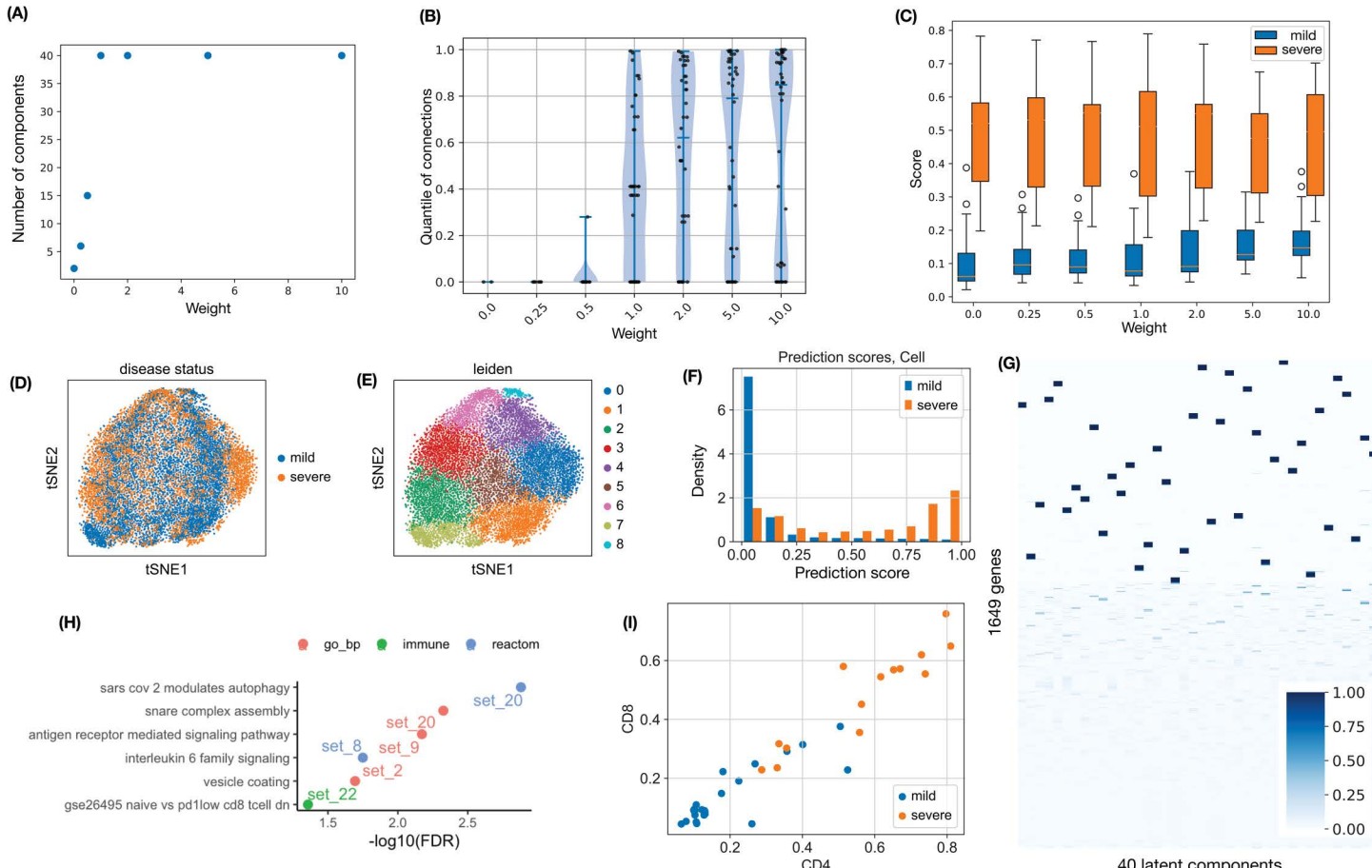

**Fig 2. Classification of severe versus mild COVID-19 patients using gene expression from CD8+T cells. (A)** Number of non-empty gene programs across different weights on the unsupervised loss. **(B)** Violin plots of connectivity quantiles. Each point represents one gene program. The connectivity quantile is computed based on the number of connections in randomly selected gene sets of the same size. **(C)** Boxplots of individual-level prediction scores (average of cell-level prediction scores) for severe disease, across different weights on the unsupervised loss. **(D–E)** t-SNE projection of CD8+T cells in the testing data, colored by disease status or by clusters identified using the Leiden algorithm. **(F)** Distribution of cell-level prediction scores (for CD8+T cells with weight 2.0) for severe disease in the testing data, stratified by the severe versus mild disease status of the corresponding individual. **(G)** Loading matrix for 1,649 genes across 40 gene programs, with genes ordered by hierarchical clustering. **(H)** Enriched functional categories of the identified gene sets. **(I)** Individual-level prediction scores for COVID-19 patients based on gene expression of CD4+ or CD8+T cells.

between 0.94 and 0.97 for both cell types (see Section A in S1 Appendix for details). Balancing predictive performance with the use of gene annotations, we selected a weight of 2 for the unsupervised loss.

Using this setting, unsupervised clustering of CD8+T cells did not clearly separate severe from mild disease status (Fig 2(D–E)). In contrast, SDAN prediction scores are very different between cells from patients with severe disease and those from patients with mild disease (Fig 2(F)). Further analysis showed that several gene programs were enriched for functional categories from gene ontology (biological processes), reactome pathways, or immunologic gene sets (Fig 2(G–H)). For example, gene program 20 was enriched for the reactome pathway of SARS-CoV-2 modulates autophagy, and GO term SNARE complex assembly. Autophagy is activated during SARS-CoV-2 infection to destroy infected cells. However, some proteins encoded by SARS-CoV-2 can prevent this process by blocking assembly of the SNARE complex, which is required for autophagy [26]. Similar enrichment patterns were observed at larger values of the unsupervised-loss weight (Fig A in S1 Appendix).

The results from CD4+T cells also supported the use of weight 2 for the unsupervised loss (Fig B in S1 Appendix). Moreover, the individual-level prediction scores derived from CD8+T cells and CD4+T cells were highly consistent (Fig 2(I)).

Finally, although individual-level prediction was based on the average of cell-level scores, the distribution of cell-level predictions is itself informative. When either CD8+T cells (Fig 2(F)) or CD4+T cells (Fig B(d) in S1 Appendix) were used to predict severe COVID-19, T cells from patients with mild disease tended to have prediction scores concentrated near 0. In contrast, T cells from patients with severe disease showed a peak near 1, but also spread in the range from 0 to 1. This pattern implies that only a subset of T cells from patients with severe disease might contribute to disease progression.

## Gene expression in astrocyte or microglia can distinguish dementia status

Alzheimer's disease, which is associated with the accumulation of amyloid beta plaques and hyperphosphorylated tau, is the most common cause of dementia in older individuals [27]. Immune processes play a central role in the pathogenesis of Alzheimer's disease [28]. Microglia and astrocyte are the two major resident immune cell types in the brain, and both are causally linked to the pathogenesis of Alzheimer's disease [28–31]. Early studies showed that activated microglia cluster around amyloid plaques [32], and later work established a causal mechanism linking microglia activity to synaptic degeneration [29]. Since then, many other studies have demonstrated the functional roles of microglia (or in general, neuroinflammation) in Alzheimer's [28], and the importance of microglia is further supported by findings from Genome-wide association studies [33]. Astrocyte have also long been established as an important neuroinflammation factor in Alizhemier's [30] and that neurotoxic astrocyte are induced by microglia [31]. Given the important and well-established causal roles of microglia and astrocyte in Alzheimer's disease, we focused on these two cell types to investigate their cell-type-specific gene expression.

We applied SDAN to single-nucleus RNA sequencing (snRNA-seq) data from the Seattle Alzheimer's Disease Cell Atlas (SEA-AD) consortium [34], focusing on ~ 70,000 astrocyte (Astro) and ~ 40,000 microglia/perivascular macrophage (Micro-PVM) cells from 84 donors, including 42 with dementia and 42 without dementia. Our goal was to classify dementia status using gene expression from astrocyte or Micro-PVM cells. As in the COVID-19 analysis, we randomly assigned half of the donors to the training set and used the remaining donors for testing. We also evaluated performance across different weights on the unsupervised loss and concluded that a value of 2.0 provided an appropriate balance (Figs C–D in S1 Appendix). We therefore focus below on the results obtained with a weight of 2.0.

Compared with the COVID-19 analysis based on T cells, predicting dementia status from astrocyte or microglia-PVM gene expression is more challenging. This can be illustrated by comparing cell-level and individual-level prediction scores in Fig 2(C, F) (CD8+T cells to predict COVID-19) and Fig 3(A–D). Many astrocyte and microglia-PVM cells have similar prediction scores between dementia and non-dementia donors (Fig 2(A–B)), though a subset of them have higher scores in dementia donors. The individual-level prediction scores only partially separate donors with dementia from those

without dementia, with AUC values of 0.744 and 0.735 for astrocyte and microglia-PVM, respectively (see Section B in S1 Appendix for details). However, when jointly considering the prediction scores by astrocyte and microglia-PVM, a subset of dementia donors shows high individual-level scores using either astrocyte or microglia-PVM (Fig 3(E)). We refer to this

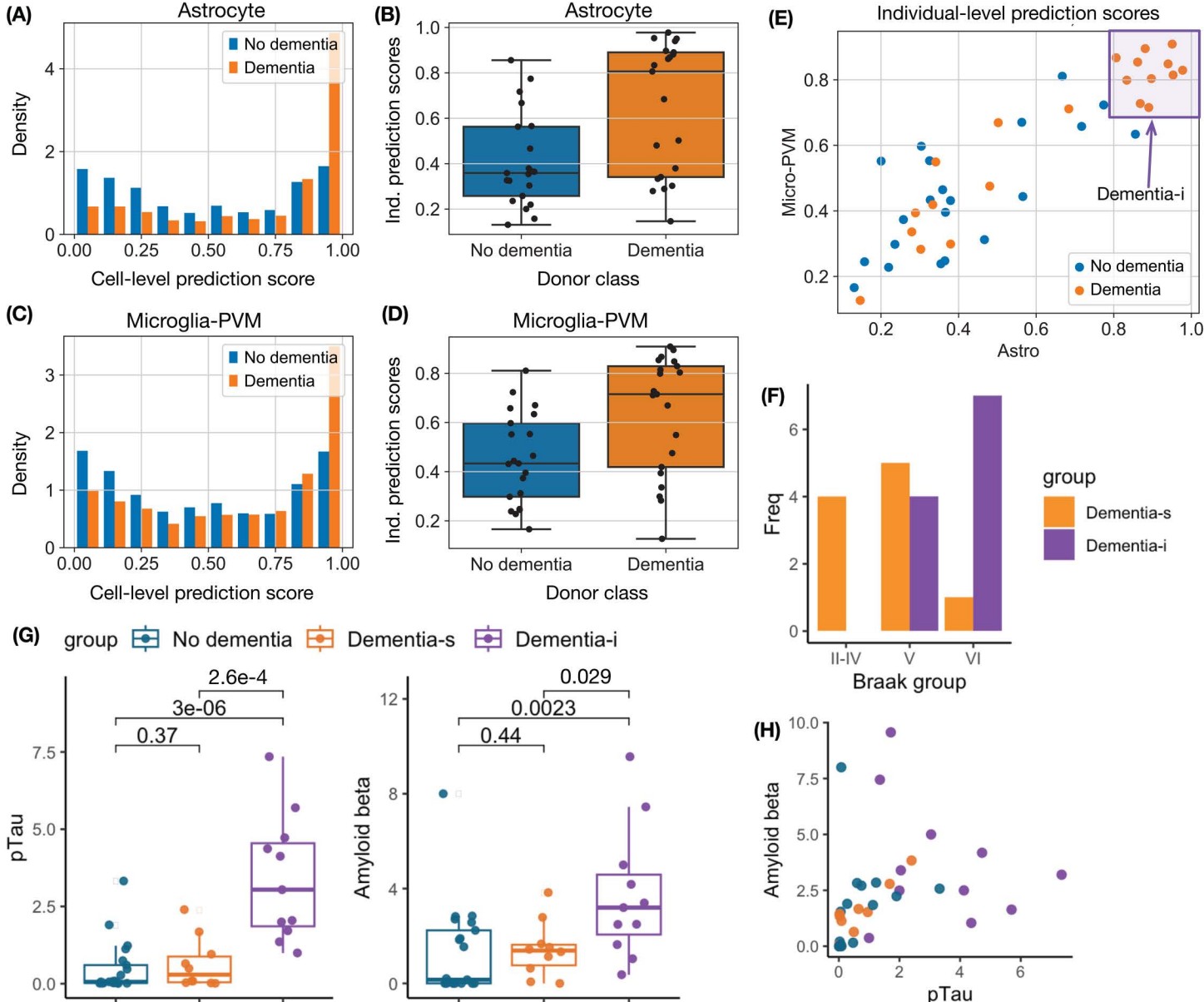

**Fig 3. Classification of dementia versus non-dementia individuals using gene expression from astrocyte and microglia-PVM cells (perivascular macrophage). (A–B)** Distributions of cell-level and individual-level prediction scores for dementia based on astrocyte in the testing data. **(C–D)** Distributions of cell-level and individual-level prediction scores for dementia based on microglia-PVM in the testing data. **(E)** Scatter plot of individual-level prediction scores by astrocyte and microglia-PVM cells in the testing data. We identified a Dementia subset. **(F)** Braak staging groups for donors in this Dementia subset and for the remaining Dementia donors. **(G)** Percentage of pTau (Amyloid beta) positive area for three donor groups: No dementia, Dementia-s, and Dementia-i. **(H)** Scatter plot of the percentage of pTau positive area versus the percentage of Amyloid beta positive area.

subset as *dementia-i*, where "i" denotes immune system, and to the remaining dementia donors as *dementia-s*, where "s" denotes immune silence.

We further characterize the dementia-i donors by the neuropathology measurements. Hyperphosphorylated tau aggregates into insoluble twisted fibers known as neurofibrillary tangles. Braak staging is a popular system used to classify the extent of neurofibrillary tangle pathology in Alzheimer's disease. The staging system is divided into six stages, with stage IV being the most severe. All four donors at Braak stage II-IV are dementia-s; most (7 out of 8) donors at Braak stage VI are dementia-i, while donors at Braak stage V are a mixture of dementia-s and dementia-i (Fig 3(F)). These results suggest that dementia-i donors have more advanced disease.

This set of dementia-i donors also has much higher pTau and amyloid beta measurements than dementia-s or non-dementia donors (Fig 3(G)), while pTau and amyloid beta were measured as the percentage of areas that are positive for pTau (phosphorylated tau, measured by antibody AT8) or Amyloid beta (by antibody 6e10) by Gabitto et al. [34]. Combining pTau and amyloid beta measurement may provide a better characterization of the dementia-i donors (Fig 3(H)), though a larger sample size is needed to confirm this conclusion.

### Gene expression in CD8+T cells can predict cancer patients' response to immunotherapy

Cancer immunotherapy, particularly immune checkpoint inhibition (ICI), has revolutionized cancer treatment, yet predicting which patients will respond remains a significant challenge [35]. The goal of ICI is to induce or strengthen the T cell response to tumors. Previous studies have demonstrated that gene expression in CD8+T cells is informative for predicting patient response to ICI [36,37]. To address this critical and challenging problem, we applied SDAN to identify gene programs associated with ICI response. Due to limited sample sizes in most existing studies, we used scRNA-seq data from one study as the training set and an independent study as the test set. Specifically, the training data came from Sade-Feldman et al. [36] and consisted of 6,350 CD8+T cells from 17 responding tumors and 31 non-responding tumors. For testing, we used the dataset from Yost et al. [37], which included 27,924 CD8+T cells from 8 responders and 7 non-responders. Notably, this analysis also evaluates the cross-study generalizability of SDAN. The model was trained on the dataset from Sade-Feldman et al. [36] and tested on the independent dataset from Yost et al. [37], which was generated in a separate study with different patients and cells. This cross-study evaluation provides a direct assessment of the robustness and transferability of SDAN when applying the learned gene-set representation to external datasets.

Direct clustering of CD8+T cells does not separate patients who respond to ICI from those who do not (Fig 4(A)). In the independent testing data from Yost et al. [37], cell-level classification performance was modest, with AUC values ranging from 0.52 to 0.59 as the unsupervised loss weights varied from 0 to 10. However, after aggregating information across cells, the individual-level AUC ranged from 0.48 to 0.82 (Section C in S1 Appendix). A weight of 2.0 for the unsupervised loss provided a reasonable balance between prediction and interpretability (Fig E in S1 Appendix). At this value, the cell-level and individual-level AUCs were 0.53 and 0.66, respectively (Fig 4(B)).

We next examined the functional categories enriched in the gene sets identified by SDAN. The top enrichment was from gene set 1, which was enriched for genes involved in mitochondrial function. This enrichment was robust across unsupervised-loss weights from 2.0 to 10 (Figs F–G in S1 Appendix). Gene set 1 contains 23 genes, including six genes that are mitochondrial ribosomal proteins or involved in mitochondrial ribosome function, three genes involved in mitochondrial transcription or translation, a mitochondrial methionyl-tRNA formyltransferase (MTFMT), two mitochondrial tRNA synthetases (SARS2 and EARS2), and four genes involved in NADH:ubiquinone oxidoreductase (Table A in S1 Appendix). NADH plays an essential role in generating energy within cells. The enrichment of mitochondrial ribosomal, tRNA synthetase, and NADH dehydrogenase genes suggests that this gene set captures mitochondrial translation and oxidative phosphorylation capacity, reflecting the metabolic fitness of T cells. Given the established role of mitochondrial function in sustaining T cell persistence and responsiveness [38–40], this gene program is likely associated with effective anti-tumor immunity and improved response to immune checkpoint blockade [41,42].

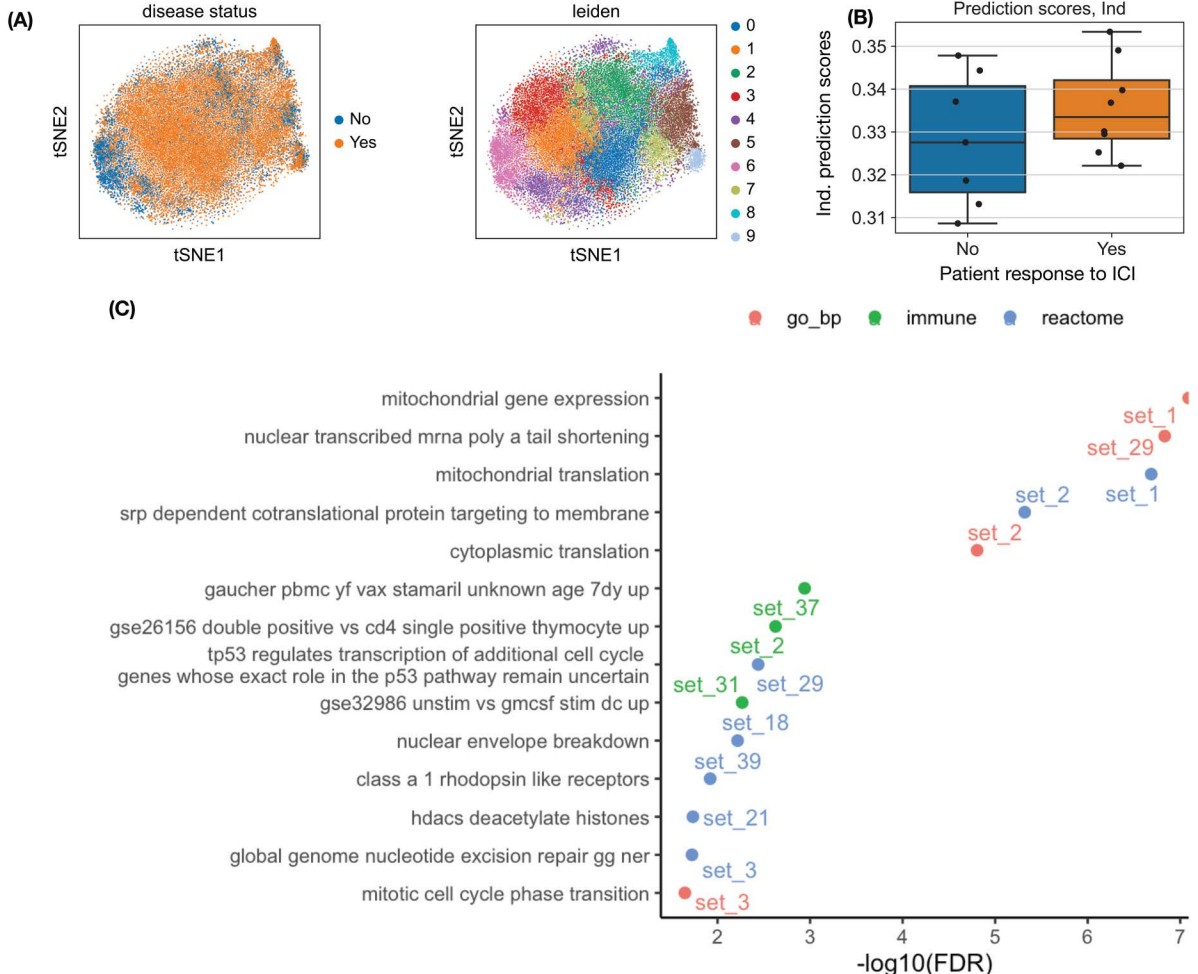

**Fig 4. Classification of cancer patients who respond to immune checkpoint inhibitors versus those who do not, using gene expression from tumor-infiltrating CD8+T cells. (A)** t-SNE projection of CD8+T cells in the testing data, colored by response to immunotherapy (Yes or No) or by clusters identified using the Leiden algorithm. **(B)** Distribution of individual-level prediction scores for patients who respond to immunotherapy (Yes) versus not (No), using an unsupervised-loss weight of 2.0. **(C)** Functional categories enriched among the 40 gene sets learned by SDAN with an unsupervised-loss weight of 2.0.

It is important to note that expression of genes located in the mitochondrial chromosome is often used as a quality-control criterion in scRNA-seq analysis, because in stressed and dying cells, the cytoplasmic mRNAs degrade faster than the mRNAs encoded by the mitochondria chromosomes. However, none of the genes in this SDAN-derived gene set is located in the mitochondrial chromosomes (Table A in S1 Appendix). Therefore, this mitochondrial signal is unlikely to reflect an underlying quality-control artifact.

## Comparison of SDAN vs. Spectra, sciRED, and scNET

We compared the performance of SDAN with three representative existing methods: Spectra [10], sciRED [9], and scNET [11]. SciRED is a representative example of factorization-based methods such as cNMF. The limitations of sciRED shown in the comparison are common features shared by all factorization-based methods: lack of sparsity in gene programs and lack of guidance from gene functional annotation.

We have compared the gene gene programs reported by SDAN, sciRED, and Spectra. Since scNET did not produce an explicit loading matrix for gene programs, we did not include it in this comparison. Because the gene programs by sciRED and Spectra were weights across many genes, we compared them with SDAN gene sets using AUC. A higher AUC indicates those genes with higher weights are more likely to belong to a SDAN gene set. For each SDAN gene set, we found its best match with the highest AUC. Many SDAN gene sets with high classification accuracies and high graph connectivities (hence better alignment of gene–gene interaction graph) did not have large overlap with any sciRED or Spectra gene programs (Figs K–L in S1 Appendix), suggesting that SDAN identify additional and informative gene sets than sciRED or Spectra.

Next, we compared their classification performances. Because Spectra, sciRED, and scNET were not originally developed as supervised learning methods, we performed a fair comparison by evaluating the gene programs identified by all four methods within a common downstream framework. Specifically, for each method, we first applied the corresponding latent-factor or representation-learning procedure to the gene expression data, then fit a logistic regression model for cell-level classification using the resulting gene-level representations, and finally aggregated the cell-level predictions to perform individual-level classification. Implementation details are provided in the Methods section.

All four methods achieved comparable performance in cell-level classification and in individual-level classification based on aggregated cell-level prediction scores (Table F in S1 Appendix). However, the gene programs identified by the four methods differed substantially in their structure and interpretability. In particular, sciRED is designed to learn gene programs without enforcing sparsity on gene loadings. As a result, each gene program typically involves thousands of genes with nonzero weights. This lack of sparsity makes the resulting gene sets difficult to interpret and less suitable for downstream biological analysis (Fig 5, Fig N in S1 Appendix).

Although the gene programs identified by Spectra enforce sparsity on gene loading vectors and therefore involve substantially fewer genes than those identified by sciRED, they remain less well defined than the gene programs inferred by SDAN (Fig 5, Figs M–O in S1 Appendix). Because both SDAN and Spectra use the same gene–gene interaction annotations to construct gene sets, we directly compared the within-gene-set connectivity induced by the annotation graph for the two methods. On average, the gene sets identified by SDAN exhibited higher connectivity in the underlying annotation graph than those identified by Spectra, indicating that SDAN more effectively recovers functionally coherent gene programs. Although scNET also uses gene–gene interaction annotations, its lack of an explicit loading matrix makes it difficult to derive directly comparable gene programs and corresponding gene sets, thereby limiting interpretability-focused comparisons with SDAN.

## Discussion

Biological knowledge, such as gene–gene interactions, is often noisy and may vary across tissues and conditions. Consequently, when incorporating such knowledge into deep learning methods, it is important to retain sufficient flexibility to learn the subset of biological information that is relevant to the specific task. SDAN addresses this challenge by combining a supervised classification loss with an unsupervised graph loss, thereby favoring gene sets that both align with biological knowledge and discriminate among cells. In contrast, many earlier studies incorporate biological knowledge into neural networks by directly constraining the network architecture using gene regulatory networks [43–46]. Such approaches are generally less flexible in selecting the subset of biological knowledge most relevant to the prediction task.

In this work, cells are labeled according to the phenotypes of the individuals from whom they were collected. For example, when comparing cells from patients with severe versus mild COVID-19, all cells from individuals with severe disease are labeled as severe, and all cells from individuals with mild disease are labeled as mild. Because of cellular heterogeneity, however, some cell-level labels inherited from individual-level phenotypes may be inaccurate. Despite this source of label noise, we show that SDAN is still able to learn useful and biologically meaningful signals. An important direction for future work is to extend SDAN to explicitly model or correct for noisy labels [47–49]. In addition, as a supervised gene

PLOS Computational Biology

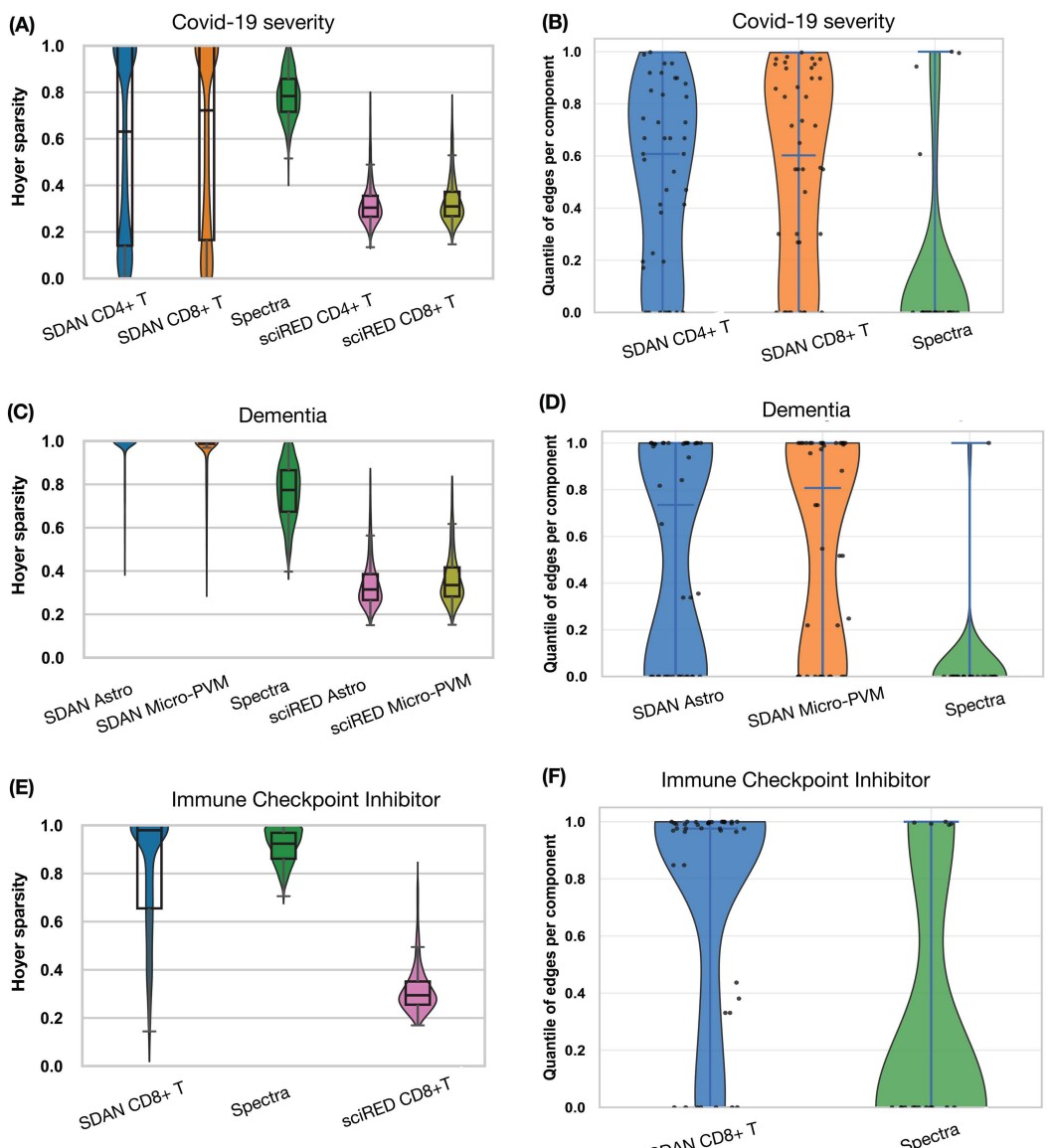

**Fig 5. Comparison of SDAN, Spectra, sciRED, and scNET. (A, C, E)** Comparison of the sparsity of gene loading vectors. We quantify the sparsity of the loading vector for each gene program using Hoyer sparsity. Hoyer sparsity is 0 when all entries are equal (maximally dense) and 1 when only one entry is nonzero (maximally sparse) (see Methods section for details). The violin plots and box plots summarize the Hoyer sparsity values across all gene programs identified by each method. Because Spectra internally accounts for cell-type specificity, its factors are not directly cell-type-specific in this comparison. scNET is not included in this analysis because it does not output an explicit gene loading vector for each gene program. **(B, D, F)** Connectivity enrichment of inferred gene sets. We assess whether genes within each inferred gene set exhibit more connections in the gene–gene interaction knowledge graph than expected by chance. A reference distribution is generated by repeatedly sampling random gene sets of the same size, and the connectivity enrichment of each gene program is quantified by its quantile within this reference distribution.

program learning method, SDAN focuses on gene programs that are associated with the phenotype of interest and will not detect gene programs that have similar activities across different phenotype values.

We compared SDAN with three alternative methods: sciRED, Spectra, and scNET. In cell-level differential expression and classification analyses, high classification accuracy is often attainable, partly because of the large number of cells

available. Consistent with this expectation, all four methods achieved similar classification accuracy across the three data-sets. However, SDAN provided superior interpretability by identifying well-defined gene sets with strong within-connectivity in known gene–gene annotation networks. Accordingly, SDAN should be viewed primarily as a method for discovering phenotype-relevant gene programs, with phenotype predictive accuracy serving as part of the loss function rather than the sole endpoint.

## Methods

### Neural network architecture and loss functions

Let $n$ denote the number of cells in the training data, $p$ the number of genes, and $d$ the number of gene programs after dimensionality reduction. Each gene is treated as a node in a gene–gene interaction graph, and the node features are given by the gene expression data. We use a graph pooling operator to reduce the gene expression representation from $p$ genes to $d$ gene programs. Let the transposed gene expression matrix be denoted by $X \in \mathbb{R}^{p \times n}$. Gene annotations are summarized by an adjacency matrix $A \in \mathbb{R}^{p \times p}$, including self-connections. Some annotations, such as protein–protein interactions, are naturally represented in adjacency matrix form. For other annotations that group genes into categories, such as Gene Ontology, gene–gene similarities can be defined as continuous measures [50], thereby yielding a weighted gene–gene interaction graph. Thus, $A$ need not be binary and may instead encode continuous similarity values. Let $\tilde{A}$ denote the normalized adjacency matrix, $\tilde{A} = D^{-1/2} A D^{-1/2}$, where $D$ is the degree matrix of $A$. The goal is to learn a loading matrix $S \in \mathbb{R}^{p \times d}$ that assigns the $p$ genes to $d$ gene programs.

The GNN consists of two steps: graph convolution followed by graph pooling. We first apply graph convolutional networks [51] to reduce the high-dimensional representation of each gene (i.e., its expression across $n$ cells) to a lower-dimensional hidden representation of size $h$:

$$H^{(1)} = \mathrm{ReLU}(\tilde{A}XW + XW_{\mathrm{skip}}),$$

where $H^{(1)} \in \mathbb{R}^{p \times h}$ is the output of the first layer, $W \in \mathbb{R}^{n \times h}$ and $W_{\mathrm{skip}} \in \mathbb{R}^{n \times h}$ are weight matrices to be estimated, and ReLU denotes the rectified linear unit activation function. Here, $\tilde{A}X$ represents the graph-convolved gene expression matrix. The same weight matrix $W$ is applied to all genes, analogous to convolutional neural networks in which the same convolutional filter is applied across different regions of an image. The additional term $XW_{\mathrm{skip}}$ allows each gene to retain greater influence from its own expression profile. This graph convolution layer can be stacked multiple times. For the $\ell$-th layer, the formulation is

$$H^{(\ell)} = \mathrm{ReLU}(\tilde{A}H^{(\ell-1)}W^{(\ell-1)} + H^{(\ell-1)}W^{(\ell-1)}_{\mathrm{skip}}).$$

The output of the final graph convolution layer is denoted by $H$. In our implementation, we used two graph convolution layers, each with 64 hidden units.

The second step is graph pooling. In the simplest setting, graph pooling is performed in a single step. The loading matrix is defined as

$$S = \mathrm{Softmax}(HW_H) \in \mathbb{R}^{p \times d},$$

where the Softmax operation is applied row-wise so that each gene is assigned a set of weights across the $d$ gene programs, and $W_H \in \mathbb{R}^{h \times d}$ is a weight matrix to be estimated. Graph pooling may also be extended to multiple layers, resulting in hierarchical graph pooling. Let $X_r = X^{\top}S$ denote the projection of the scRNA-seq data into the latent space. We use $X_r$ to predict class labels via a multilayer perceptron (MLP) with three hidden layers, each containing 64 hidden units and a ReLU activation function.

We define the loss function as the sum of two components: an unsupervised graph loss and a supervised classification loss. For the graph loss, we adopt the minCUT loss, which is designed to resemble spectral clustering [52]. The graph loss consists of two terms, $\mathcal{L}_c$ and $\mathcal{L}_o$, defined as

$$\mathcal{L}_c := -\frac{\mathrm{Tr}(S^\top \tilde{A} S)}{\mathrm{Tr}(S^\top \tilde{D} S)} \quad \text{and} \quad \mathcal{L}_o := \left\| \frac{S^\top S}{\|S^\top S\|_F} - \frac{I_d}{\sqrt{d}} \right\|_F,$$

where $\|\cdot\|_F$ denotes the Frobenius norm, $\tilde{A}$ is the normalized adjacency matrix, $\tilde{D}$ is the degree matrix of $\tilde{A}$, and $I_d$ is the $d$-dimensional identity matrix. The term $\mathcal{L}_c$ encourages strongly connected nodes to be grouped together. The term $\mathcal{L}_o$ encourages the gene loading vectors to be approximately orthogonal, thereby promoting assignment of each gene to a single gene program and encouraging the gene programs to have comparable sizes. The classification loss is given by the cross-entropy loss, denoted by $\mathcal{L}_{clf}$. The final loss function is therefore

$$\mathcal{L}_{clf} + w(\mathcal{L}_c + \mathcal{L}_o),$$

where $w$ controls the strength of the unsupervised graph regularization. We set $w = 2$ by default.

In practice, $w$ should be selected by balancing predictive performance and interpretability. We recommend fitting SDAN over a grid of values, e.g., $w \in \{0, 0.25, 0.5, 1, 2, 5, 10\}$, and evaluating each fit using both validation performance and structural diagnostics of the inferred gene programs. Useful diagnostics include the within-program connectivity quantile relative to randomly selected gene sets of the same size, the number of non-empty gene programs, and the functional enrichment of the inferred gene sets. We recommend selecting the smallest $w$ for which validation performance remains close to the best observed value and these graph-based diagnostics have stabilized. When phenotype labels are available but potentially noisy, the absolute level of prediction accuracy may be limited by label uncertainty. In this setting, however, validation performance still provides useful relative information across values of $w$, especially when interpreted together with the graph-based diagnostics described above.

A key advantage of SDAN is that the annotation graph is incorporated through a weighted graph-regularization term rather than being hard-coded into the model structure. As a result, SDAN can adaptively balance biological interpretability and predictive accuracy: when the prior is informative, it encourages sparse and functionally coherent gene programs, whereas when the prior is less informative, the supervised objective can still preserve classification performance. This flexibility distinguishes SDAN from methods that ignore prior knowledge and from approaches that rely more rigidly on predefined biological networks.

To assess robustness to the annotation graph, we performed a graph-sensitivity analysis in which the annotation graph was progressively perturbed while keeping the data, selected genes, and model hyperparameters fixed. These results are reported in Section D.1 in S1 Appendix and show that test AUC remains relatively stable even under substantial graph perturbation, indicating that SDAN's predictive performance is not overly dependent on the exact annotation graph. In contrast, the structure of the learned loading matrix is more sensitive to graph quality: as the graph is increasingly perturbed, the learned gene programs become less sparse, less well separated, and more overlapping, indicating reduced interpretability. These findings suggest that a biologically meaningful annotation graph mainly improves the coherence and interpretability of the learned gene programs, whereas prediction remains comparatively robust even when the prior is degraded.

## Evaluation and generalization

A key advantage of SDAN is that the learned loading matrix $S$, which maps genes to gene programs, is independent of individual cells and can therefore be reused across datasets. When applying the model to a new dataset, we align the

gene lists between the training and test data and project the test data into the learned latent space using the corresponding submatrix of $S$. This design allows SDAN to generalize to datasets containing different cells and potentially different, but overlapping, gene sets.

Let $X_{\text{test}} \in \mathbb{R}^{n' \times p'}$ denote the test data, where $p'$ need not equal $p$; that is, the gene lists in the training and test data may differ. If the gene list in the test data is identical to that in the training data, dimensionality reduction is performed as

$$X_{\text{test,r}} = X_{\text{test}} S,$$

using the trained loading matrix $S$. We then apply the trained MLP to $X_{\text{test,r}}$ to obtain a prediction score for each cell in the test data. If the gene list in the test data differs from that in the training data, we first identify the genes shared between the two datasets. We then use the corresponding submatrices of $X_{\text{test}}$ and $S$ to project the test data into the learned latent space.

## Extensions

Although the applications considered in this paper focus on binary phenotypes, the SDAN framework is not restricted to binary outcomes. SDAN is formulated as a supervised representation-learning framework in which phenotype information enters through the supervised loss term. For multi-class phenotypes, the binary cross-entropy loss can be replaced with the standard softmax cross-entropy loss, with the classifier outputting a probability vector over multiple classes. For continuous outcomes, the supervised objective can be replaced with a regression loss such as mean squared error. In both cases, the graph loss used to learn gene programs remains unchanged. Aggregation from cell-level predictions to the individual level can be performed using the same strategy as in the binary setting, for example by averaging predicted scores or probabilities across cells from the same individual. This flexibility allows SDAN to be applied to a broad range of phenotype types while preserving the interpretability of the learned gene programs.

In the current implementation, SDAN is applied to one cell type at a time. When the scientific objective is to integrate information across multiple cell types, one natural extension is to fit separate SDAN models for each cell type, aggregate cell-level prediction scores within each donor and cell type, and then combine these donor-level scores using a downstream meta-classifier. Alternatively, SDAN can be extended to a multi-branch architecture with one cell-type-specific branch for each cell type, followed by a shared donor-level integration layer that is optimized jointly across branches. These extensions would preserve the interpretability of cell-type-specific gene programs while enabling joint prediction based on complementary signals from multiple cell types.

## Evaluation metrics

To quantify the sparsity of the learned loading vector for each gene, we use *Hoyer sparsity*, a scale-invariant measure of how strongly a vector is concentrated on a small number of entries. For a vector $\mathbf{v} \in \mathbb{R}^d$, Hoyer sparsity is defined in terms of its $L_1$ and $L_2$ norms as

$$\text{Sparsity}(\mathbf{v}) = \frac{\sqrt{d} - \|\mathbf{v}\|_1 / \|\mathbf{v}\|_2}{\sqrt{d} - 1},$$

which takes values in [0,1]. Values close to 0 correspond to dense vectors with relatively uniform weights, whereas values close to 1 indicate highly sparse vectors dominated by a small number of nonzero entries. Importantly, Hoyer sparsity is invariant to rescaling of the vector, making it well suited for comparing sparsity across methods with different normalization schemes. In our analysis, we compute Hoyer sparsity for each gene using its loading vector across gene programs in the

learned loading matrix. Higher Hoyer sparsity therefore indicates a more interpretable representation, as the gene loads on only a small number of gene programs, or even a single gene program.

## Implementation of SDAN

The gene–gene interaction graph was constructed using protein–protein interaction annotations from the BioGRID database, specifically the file `BIOGRID-ORGANISM-Homo_sapiens-4.4.204.tab3.txt.gz`. Genes were matched by their official gene symbols in BioGRID. Although BioGRID records interactions between proteins, each protein entry is linked to a corresponding gene symbol. In cases where multiple protein isoforms corresponded to the same gene, these isoforms were collapsed into a single gene-level node. An edge was placed between two genes if an interaction between any of their encoded proteins was recorded in the database. Thus, the final graph is a gene-level interaction network derived from protein-level evidence.

Given cell-level labels (e.g., severe versus mild COVID-19, dementia versus no dementia, or response versus non-response to immunotherapy), we first performed differential expression (DE) analysis to identify candidate genes. This step serves as a preselection procedure to reduce the original high-dimensional gene space to a manageable size for model training. Specifically, we identified marker genes separately for each class label. For example, in the severe versus mild COVID-19 setting, we performed a one-sided Mann–Whitney $U$ test for each gene using the training data. To identify marker genes for the severe (or mild) group, the alternative hypothesis was that expression in that group was stochastically greater than in the other group. This procedure generalizes naturally to multi-class settings by comparing one class against all remaining classes. To account for multiple testing, we controlled the false discovery rate (FDR) using the Benjamini–Hochberg procedure and retained genes with FDR $\leq 0.05$. Because the number of selected marker genes could still be large, we further restricted the set by retaining at most 1,000 highly variable marker genes per class, ranked by normalized variance computed from normalized expression data across all class labels. The marker genes selected for all classes were then combined to form the gene list used for model fitting.

We emphasize that this DE analysis is used only as a preselection step to reduce the original high-dimensional gene space to a manageable size for model training. The preselection is intentionally inclusive: after identifying DE genes for each class, we retain up to 1,000 highly variable marker genes per class and merge these sets across classes to construct a relatively large candidate gene pool. The final gene programs and gene sets are then learned by SDAN itself. Accordingly, genes that pass this initial filter may still receive negligible weights in the learned gene programs. To assess the sensitivity of SDAN to this preselection step, we varied the FDR threshold used for DE-based gene selection and reran SDAN under each setting. The results indicate that, even when more genes are selected using more liberal DE thresholds, a similar number of genes are ultimately included in the SDAN gene programs and the results remain robust to the DE threshold (Section D.2 in S1 Appendix).

For dimensionality reduction, we set the number of gene programs to 40. To define the genes corresponding to each gene program, we classified a gene as belonging to a gene program if its loading in the matrix $S$ for that program exceeded 0.8. For the unsupervised clustering analysis based on the Leiden algorithm, which was used to evaluate the gene programs, we set the number of neighbors to 20 when constructing the neighborhood graph and used a resolution parameter of 1.

The method was implemented in PyTorch. Model parameters were optimized using the Adam optimizer with a learning rate of 0.0001 and weight decay of 0.0001. To reduce overfitting, we used both dropout and early stopping. Dropout was applied after each hidden layer of the MLP, but not after the output layer, with dropout probability 0.5. We set the maximum number of training epochs to 50,000 and the minimum number to 10,000. Training was stopped if the validation loss failed to improve for 3,000 epochs.

For individual-level prediction, we defined each individual's prediction score as the mean of the cell-level prediction scores across all cells from that individual.

## Implementation of sciRED, Spectra, and scNET

We implemented sciRED following the framework of Pouyabahar et al. [9]. After selecting DE genes using the same procedure as for SDAN, we extracted the corresponding raw count matrix. Because sciRED explicitly models technical noise, no read-depth normalization or log transformation was applied before model fitting. For each gene, we fitted a Poisson generalized linear model with an intercept, the cell-specific library size, and protocol indicators (when available) as covariates. This procedure yielded estimated mean expression values and Pearson residuals, where the residual matrix provides a variance-stabilized representation of gene expression that adjusts for sequencing depth and batch effects. The residual matrix was converted to a dense array and standardized gene-wise to have mean zero and unit variance. We then applied principal component analysis to the residuals, retaining 40 components to match the number of gene programs used in SDAN. To enhance interpretability, we applied varimax rotation to the loading matrix, yielding approximately orthogonal gene programs. The resulting cell-level latent representations were used for downstream cell-level prediction, and individual-level scores were obtained by averaging cell-level prediction scores within each individual.

We implemented Spectra following the framework of Kunes et al. [10]. When multiple cell types were present (e.g., CD4$^+$ T cells and CD8$^+$ T cells in the COVID-19 severity study), we first constructed a union gene list by combining the cell-type-specific DE genes. Spectra was then trained on the combined dataset containing cells from all relevant cell types using this union gene list. The gene expression matrix was normalized for read depth and log-transformed, consistent with the preprocessing used for SDAN, and the same gene annotation information derived from BioGRID interactions was used. Spectra decomposes the expression matrix into two types of latent factors: global factors shared across cell types and cell-type-specific factors. In our applications, there were either one or two cell types of interest. When there was only one cell type, we used 40 latent factors to match the number used in SDAN. When there were two cell types, we used 20 global factors and 10 cell-type-specific factors for each cell type, following the model structure suggested by Kunes et al. [10]. After fitting Spectra, we obtained a loading matrix mapping genes to gene programs. We then projected the read-depth-normalized and log-transformed gene expression data into the latent space using the learned loading matrix. These latent representations were used for downstream cell-level prediction, and individual-level scores were obtained by averaging cell-level prediction scores within each individual.

We implemented scNET following the graph-based representation learning framework of Sheinin et al. [11]. After selecting DE genes using the same procedure as for SDAN, we extracted the corresponding gene expression matrix and constructed the same gene annotation graph based on BioGRID interactions. To reduce computational cost, we concatenated the training and test cells and performed balanced subsampling to retain up to 7,500 cells from each partition, following the strategy suggested by Sheinin et al. [11]. The resulting gene expression matrix was converted to a dense matrix and standardized gene-wise to have mean zero and unit variance. In parallel, we constructed a cell–cell $k$-nearest neighbor graph using 10 neighbors and 15 principal components. scNET was then trained jointly on the gene annotation graph, the cell–cell $k$-nearest neighbor graph, and the standardized gene expression matrix using a graph neural network model with the default embedding dimensions specified in [11]. After training, we extracted the learned cell embeddings and used them as latent representations for downstream cell-level prediction; individual-level scores were obtained by averaging cell-level prediction scores within each individual. Unlike SDAN, sciRED, and Spectra, scNET does not produce an explicit gene loading matrix for gene programs, making its learned representations not directly interpretable as gene programs with corresponding gene sets.

## Supporting information

**S1 Appendix. Supplementary methods and results.**
(PDF)

## Author contributions

**Conceptualization:** Wei Sun.

**Data curation:** Zhexiao Lin, Wei Sun.

**Formal analysis:** Zhexiao Lin, Yuanyuan Gao.

**Funding acquisition:** Wei Sun.

**Investigation:** Wei Sun.

**Methodology:** Zhexiao Lin, Yuanyuan Gao, Wei Sun.

**Project administration:** Wei Sun.

**Resources:** Wei Sun.

**Software:** Zhexiao Lin, Yuanyuan Gao.

**Supervision:** Wei Sun.

**Validation:** Zhexiao Lin, Yuanyuan Gao.

**Visualization:** Zhexiao Lin, Yuanyuan Gao, Wei Sun.

**Writing – original draft:** Wei Sun.

**Writing – review & editing:** Zhexiao Lin, Yuanyuan Gao.

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
