## [Decision Letter · Decision Letter 0]

1 Mar 2026

PCOMPBIOL-D-26-00188

Supervised deep learning with gene annotation for cell classification

PLOS Computational Biology

Dear Dr. Sun,

Thank you for submitting your manuscript to PLOS Computational Biology. After careful consideration, we feel that it has merit but does not fully meet PLOS Computational Biology's publication criteria as it currently stands. Therefore, we invite you to submit a revised version of the manuscript that addresses the points raised during the review process.

We look forward to receiving your revised manuscript.

Kind regards,

Peng Wei, Ph.D.

Academic Editor

PLOS Computational Biology

Marc Birtwistle

Section Editor

PLOS Computational Biology

**Additional Editor Comments:**

Your manuscript has been assessed by our reviewers who are experts in the field. Based upon these reviews, we would be willing to consider a revision. You would need to compare and evaluate the proposed method with both standard benchmark methods and more recently proposed method such as scNET. In addition, please address reviewers’ comments on clarification of the primary objective, filtering prior to GNN, and interpretation of real data analysis results.

**Journal Requirements:**

At this stage, the following Authors/Authors require contributions: Yuanyuan Gao, Zhexiao Lin, and Wei Sun. Please ensure that the full contributions of each author are acknowledged in the "Add/Edit/Remove Authors" section of our submission form.

4) Please amend your detailed Financial Disclosure statement. This is published with the article. It must therefore be completed in full sentences and contain the exact wording you wish to be published.

5) Please send a completed 'Competing Interests' statement, including any COIs declared by your co-authors. If you have no competing interests to declare, please state "The authors have declared that no competing interests exist". Otherwise please declare all competing interests beginning with the statement "I have read the journal's policy and the authors of this manuscript have the following competing interests"

**Reviewers' comments:**

Reviewer's Responses to Questions

**Comments to the Authors:**

Reviewer #1: The manuscript introduces Supervised Deep learning with gene ANnotation (SDAN), a graph neural network (GNN)-based framework designed for the supervised classification of cells and individuals using single-cell RNA-sequencing (scRNA-seq) data. The core motivation is to move beyond traditional gene-by-gene differential expression (DE) analysis—which often yields long, uninterpretable lists of genes with small effect sizes—toward identifying functionally coherent gene sets that directly optimize classification performance. SDAN integrates gene expression profiles with gene-gene interaction networks (e.g., BioGRID) and employs a graph pooling operation to cluster genes into latent components. These components serve as "gene sets" used to classify cell states (e.g., severe vs. mild COVID-19) and are subsequently aggregated to provide individual-level clinical predictions. My comments are listed below:

Major:

1. SDAN has a similar design as scNET (Ron Sheinin et al. Nature Methods. 2025): both using scRNA-seq data and gene-gene interaction as input into a graph neural network for better functional annotation of scRNA-seq across cell types or biological conditions. The authors should compare scNET with SDAN, and discuss the strengths and weaknesses of each methods.

2. The authors state that they pre-select marker genes using a one-sided Mann-Whitney U test to facilitate training. While this reduces computational overhead, it may introduce a "winner's curse" where the model only builds gene sets around genes that already show strong individual separation. A key promise of GNNs is the ability to pick up on subtle, coordinated signals from connected nodes. The authors should discuss whether SDAN can identify meaningful gene sets if pre-selection is relaxed, or provide a sensitivity analysis on how the pre-selection threshold affects the discovery of novel biology.

3. While the source code for SDAN is provided on GitHub, I encountered difficulties in creating a functional runtime environment due to unresolved dependency conflicts. To ensure the method is accessible to the broader research community, I strongly recommend that the authors distribute SDAN as a formal Python package (e.g., installable via pip or conda). At a minimum, the authors must provide a comprehensive requirements.txt or environment.yml file with strictly pinned versions for all dependencies to ensure long-term reproducibility.

Minor:

1. The term “gene annotation” is inappropriately used. According to Ensembl, Gene annotation is the plotting of genes onto genome assemblies, and indexing their genomic coordinates (http://useast.ensembl.org/info/genome/genebuild/index.html). In the manuscript, it will be better described as “gene-gene interaction”, “protein-protein interaction”, or similar terms.

2. For figure 5, “Hoyer sparsity” is described in the legend, but “hyper sparsity” is shown in the figure. Are they the same thing? In addition, the evaluation metrics, including Hoyer sparsity and quantiles of edges per component, should be clearly defined in the main Methods section, and explained why these metrics indicate better performance.

3. The authors compared SDAN, Spectra, and sciRED across multiple datasets. It would be beneficial to include a direct comparison of the "functionally coherent" gene sets found by each method for the same biological process (e.g., show the gene weights of a biological pathway known to be involved in COVID-19, dementia or immunotherapy in all three methods) to demonstrate the "unambiguous assignment" claimed in the abstract.

Reviewer #2: Please see the attachment for review.

Reviewer #3: This study presents SDAN, a computational framework which incorporates graph neural network to learn gene set assignments for the classification of cell-level and patient-level metadata. While the manuscript evaluates SDAN on three datasets and benchmarks it against two existing methods, the presentation of the results could be strengthened to more clearly highlight the model’s novelty and its performance advantages. Below are my detailed comments.

Major Comments:

1. The main motivation of this work is to solve the effect size issue of differential expression (DE) analysis arising from the large sample size typical of scRNA-seq data, where even genes with minimal biological effect can achieve extremely small p-values. To mitigate this, the study proposes learning gene set assignments based on the top DE genes identified. While this is an interesting direction, it would be helpful to more clearly explain how this strategy substantively mitigates the stated issue. Specifically, because genes are initially selected based on statistical significance, genes with negligible biological relevance may still be maintained. Conversely, genes that do not meet the initial DE threshold are excluded entirely and cannot be reconsidered later, potentially introducing an irreversible selection bias. It would strengthen the manuscript to provide additional justification or empirical evidence demonstrating that the proposed framework meaningfully alleviates the effect size concern.

2. Figure 2C&F, 3A-B & C_D show the cell-level and individual-level precision score respectively. However, Figure 2F and 3A&B’s x-axis and y-axis is not labeled, which makes interpretation difficult. In addition, only the distribution of prediction scores is shown, while the classification performance is not presented. It would be more informative to report the classification metrics, such as AUC score, of SDAN and compare it to the benchmarking methods.

3. The study reports gene set enrichment analysis results to demonstrate the biological relevance of the identified gene sets, it remains unclear how these gene sets are intended to replace or improve upon traditional DE analysis. The manuscript would benefit from a clearer explanation of how gene set–level outputs provide comparable or superior interpretability relative to standard DE results. Additionally, some gene sets may contribute to both classes. For example, mild and severe condition of COVID-19 may share similar pathway, but the degree of contribution can be different. Clarifying how SDAN captures and quantifies such differential contributions would enhance the interpretability claims of the framework.

4. An interesting aspect of SDAN is that the learned mapping matrix S is cell independent. In principle, this suggests that the model could be applied to other datasets that share the same gene measurements. Beyond evaluating performance through a train–test split within the same dataset, it would be valuable to assess the generalizability of SDAN on more biologically heterogeneous datasets, such as across independent studies. Such analysis would provide stronger evidence of the robustness and transferability of the learned representation.

5. In lines 171-173, the manuscript states that “Combining pTau and amyloid beta measurement may give a better characterization of the dementia-i donors (Fig 3(H)), though a larger sample size is needed to confirm this conclusion.” Does the current modeling framework allow for the integration of two or more cell types simultaneously, and if so, how would it be extended to accommodate such multi-cell-type inputs?

Minor comments

1. The gene-gene network is constructed from PPI. It is unclear how this information is mapped or utilized at the gene level in the proposed framework. Clarification on this point would be helpful.

**Have the authors made all data and (if applicable) computational code underlying the findings in their manuscript fully available?**

Reviewer #1: Yes

Reviewer #2: Yes

Reviewer #3: Yes

PLOS authors have the option to publish the peer review history of their article (what does this mean?). If published, this will include your full peer review and any attached files.

**Do you want your identity to be public for this peer review?** For information about this choice, including consent withdrawal, please see our Privacy Policy.

Reviewer #1: No

Reviewer #2: No

Reviewer #3: No

**Figure resubmission:**
---

## [Decision Letter · Decision Letter 1]

12 May 2026

Dear Dr. Sun,

We are pleased to inform you that your manuscript 'Supervised deep learning with gene annotation for cell classification' has been provisionally accepted for publication in PLOS Computational Biology.

Best regards,

Peng Wei, Ph.D.

Academic Editor

PLOS Computational Biology

Marc Birtwistle

Section Editor

PLOS Computational Biology

We appreciate the authors' efforts in addressing the reviewers' comments. Reviewer 1 has some minor suggestion regarding the use of "gene function". Please consider changing it if possible.

Reviewer's Responses to Questions

**Comments to the Authors:**

Reviewer #1: The authors have addressed most of my comments. I suggest changing “gene annotation” to either “gene functional annotation”, “functional annotation” or “gene-gene interaction” throughout the manuscript depending on the context. This avoid the confusion with the gene structural annotation, which is also commonly known as “gene annotation” in genetics and bioinformatics studies.

Reviewer #2: The authors have addressed my questions.

Reviewer #3: Thank you for your revised manuscript and detailed response letter addressing my previous comments. I appreciate the efforts made to improve the manuscript. I have no additional comments.

**Have the authors made all data and (if applicable) computational code underlying the findings in their manuscript fully available?**

Reviewer #1: Yes

Reviewer #2: None

Reviewer #3: Yes

PLOS authors have the option to publish the peer review history of their article (what does this mean?). If published, this will include your full peer review and any attached files.

Reviewer #1: No

Reviewer #2: No

Reviewer #3: No

---

## [Editor Report · Acceptance letter]

PCOMPBIOL-D-26-00188R1

Supervised deep learning with gene annotation for cell classification

Dear Dr Sun,

I am pleased to inform you that your manuscript has been formally accepted for publication in PLOS Computational Biology. Your manuscript is now with our production department and you will be notified of the publication date in due course.

With kind regards,

Anita Estes
